# Leveraging Language for Accelerated Learning of Tool Manipulation

**Allen Z. Ren**[1], **Bharat Govil**[2], **Tsung-Yen Yang**[2], **Karthik Narasimhan**[2]*, **Anirudha Majumdar**[1]*

[1]Department of Mechanical and Aerospace Engineering, [2]Department of Computer Science
Princeton University
{allen.ren, bgovil, ty3, karthikn, ani.majumdar}@princeton.edu

**Abstract:** Robust and generalized tool manipulation requires an understanding of the properties and affordances of different tools. We investigate whether linguistic information about a tool (*e.g.,* its geometry, common uses) can help control policies adapt faster to new tools for a given task. We obtain diverse descriptions of various tools in natural language and use pre-trained language models to generate their feature representations. We then perform language-conditioned meta-learning to learn policies that can efficiently adapt to new tools given their corresponding text descriptions. Our results demonstrate that combining linguistic information and meta-learning significantly accelerates tool learning in several manipulation tasks including pushing, lifting, sweeping, and hammering. [2]

**Keywords:** Language for Robotics, Tool Manipulation, Meta-learning

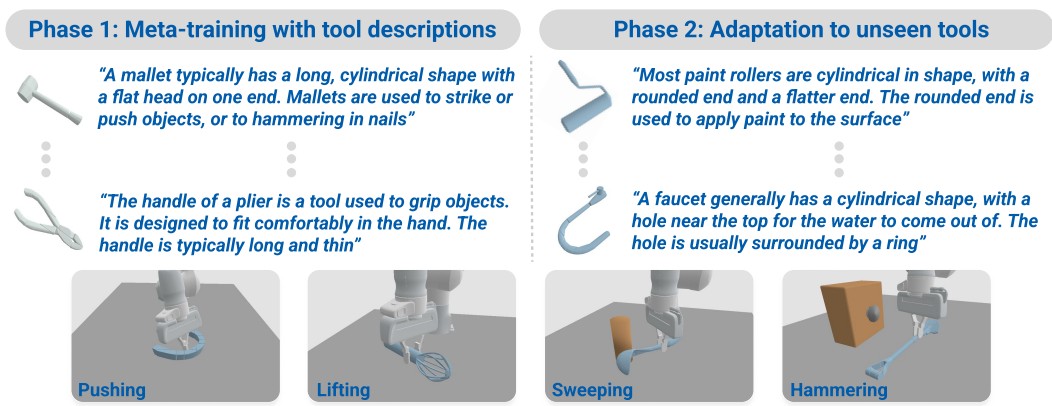

Figure 1: Rich, semantic knowledge from language descriptions, such as geometric features and common use of the tools, can help policies adapt faster to unseen tools (right) in pushing, lifting, sweeping, and hammering tasks (bottom) after meta-learning on training tools (left).

## 1 Introduction

The ability to quickly learn how to use a broad range of new tools is considered one of the hallmarks of human intelligence [1]. Humans are able to attain a degree of proficiency with a new tool (*e.g.,* a new hand tool or a virtual tool such as a joystick or remote) within just a few minutes of interaction [2]. This rapid learning relies on the ability to understand or discover the *affordances* [3] of a new tool, *i.e.,* the ability to perform a certain action with the tool in a given environment. For example, a hammer affords the opportunity to grasp it, use it to hammer a nail, or use it to push another object. In contrast, a spatula affords the opportunity to flip a pancake or sweep food ingredients into a bowl. Endowing robots with the ability to quickly discover and exploit affordances of a new tool in order to learn how to perform a given task has been a long-standing grand challenge in robotics [4].

---

*Equal contribution in advising
[2]Video showing the four manipulation tasks: https://shorturl.at/dmCEW

6th Conference on Robot Learning (CoRL 2022), Auckland, New Zealand.

While humans routinely rely on geometric priors and visual observations for tool manipulation, they can also 'read' text manuals or linguistic descriptions to understand affordances of new tools and quickly adapt to using them. In this work, we investigate whether *language* can help robots accelerate the process of learning to use a new tool. In particular, consider the following descriptions of two types of tools:

> *The shape of tongs is typically that of a V, with two long, thin handles that come to a point at the top, and a gripping area in the middle.*
>
> *A spatula is a kitchen utensil for flipping over food while cooking. The head of the spatula is usually rectangular or oval in shape. The handle of the spatula is usually long and thin.*

Our key intuition is that such natural language descriptions of tools contain information about the affordances of the tools, how to exploit these affordances for a given task, and how perceptual features of tools (*e.g.,* their visual appearance and geometry) relate to their affordances. Moreover, language can help capture the *shared structure* of tools and their affordances. Thus, if one has previously learned to use a set of tools (with corresponding language descriptions), a description of a new tool can help to exploit this prior knowledge in order to accelerate learning.

**Statement of Contributions.** Our primary contribution is to propose **ATLA** — **A**ccelerated Learning of **T**ool Manipulation with **LA**nguage — a meta-learning framework leveraging large language models (LLMs) to accelerate learning of tool manipulation skills. The overall approach is illustrated in Fig. 1. We propose to use LLMs in two distinct ways: to generate the language descriptions for tools and to obtain the corresponding feature representations. At meta-training time, the meta-learner updates a base-learner that quickly fine-tunes a manipulation policy; this fine-tuning process is conditioned on the LLM representations corresponding to the language descriptions of each tool. Specifically, we propose a simple gradient-based meta-learning setup based on Reptile [5] that performs off-policy updates. At test time, the base-learner adapts to a new tool using its language descriptions and interactions with it. To our knowledge, our approach is the first to utilize LLMs to accelerate learning of new tools. We demonstrate the benefits of using language in a diverse set of tool-use tasks including pushing, lifting, sweeping, and hammering.

## 2 Related Work

**Tool Manipulation.** Tool manipulation [6, 7, 8, 9, 10] is one of the long-standing problems in robotics research. A major challenge is understanding the affordances of the tool in different tasks. Previous work has modeled and learned affordances from parameterized keypoints on the tools [7, 10], from human demonstrations [8], and from spatial-temporal parsed graphs of the tools [11]. Our work instead leverages natural language (*e.g.,* describing affordances of the tools in words) for generalization of affordances in tool manipulation and is compatible with previous approaches.

**Language-informed Control.** Natural language has been applied to enable efficient robotic learning through (1) generating primitive language instructions for producing control actions (*i.e.,* instruction following task) [12, 13, 14, 15, 16, 17, 18, 19], (2) learning language-informed reward functions for training control policies [20, 21, 22, 23, 24, 25], and (3) using language to correct or adapt the behavior of the robot [26, 27]. However, these works primarily translate natural language into action policies for a specific task with the text providing information on the desired actions that optimize returns (*e.g.,* "*push the door*"). This means that the text is tightly coupled with the task seen during training, making it difficult to generalize to a new distribution of tasks with different dynamics. In contrast, the text in our work only provides a high-level description of the property of each tool, encouraging the agent to extract useful information to generalize to a new task. Some prior work [28, 29, 30, 31] also uses language descriptions of environment dynamics to enable generalization of policies but does not leverage meta-learning.

**Meta-learning.** Our work uses the framework of meta-learning [32, 33, 34, 35, 36], in which the agent is trained with a distribution of tasks, and later adapts quickly to a previously unseen task. "Reptile" is proposed in [5] as a simple first-order, gradient-based meta-learning algorithm that learns an initialization of the neural network's parameters for fast adaptation at test time. Recent papers [37, 38, 39, 40] have also explored providing additional context information (*e.g.,* the property of the task) to encode task-specific knowledge for a meta-learning agent. However, all these works directly provide the context information either through scalar signals or a learned task embedding, which require domain expertise or a pre-training stage. In this work, we assume that the agent is provided with a text description of the tool, which is more accessible and easier to collect.

# 3 Problem Formulation

We consider the following goal: given a new tool and corresponding language description(s), we aim to *quickly* learn a policy for using the tool to achieve a given task. We pose this problem in a meta-learning setting in which a policy is trained with a distribution of tools for a given task, and later adapts quickly to a previously unseen tool sampled from this distribution in the same task.

**Meta-training.** During meta-training, we assume access to a set $\mathcal{T} = \{\tau_i\}_{i=1}^{K}$ of tools, where $K$ is the number of available tools. For each tool $\tau_i$, we are also provided a set of corresponding language descriptions $L_i = \{l_{ij}\}_{j=1}^{N_i}, l_{ij} \in \mathcal{L}$, where $N_i$ is the number of available descriptions for tool $\tau_i$ and $\mathcal{L}$ is the set of possible textual descriptions. In addition, each $l_{ij}$ can describe a different property of tools such as shape and common use. Given a particular robotic manipulator and a particular task (*e.g.,* pushing, lifting, sweeping, or hammering), each tool $\tau$ induces a partially-observable Markov decision process (POMDP): $\langle \mathcal{S}_\tau, \mathcal{A}, \mathcal{O}, \mathcal{P}_\tau, R_\tau \rangle$. Here, $\mathcal{S}_\tau$ is the state of the entire environment (*i.e.,* combined state of the robot, tool, and potentially other objects to be manipulated using the tool). The robot's action space $\mathcal{A}$ (*e.g.,* corresponding to robot joint torques) and observation space (*e.g.,* the space of RGB-D observations from a camera) are fixed across tools. The transition probabilities are given by $\mathcal{P}_\tau : \mathcal{S}_\tau \times \mathcal{A} \times \mathcal{S}_\tau \rightarrow [0, 1]$, and the reward function is $R : \mathcal{S}_\tau \times \mathcal{A} \times \mathcal{S}_\tau \rightarrow [0, 1]$. During meta-training, our goal is to learn a policy $\pi_\theta : \mathcal{O} \times \mathcal{L} \rightarrow \mathcal{A}$ parameterized by $\theta$ (*e.g.,* weights of a neural network) that can be quickly fine-tuned at test time.

**Meta-testing.** At test time, we are provided a new tool $\tau_\nu$ and corresponding language descriptions $L_\nu = \{l_{\nu j}\}_{j=1}^{N_\nu}, l_{\nu j} \in \mathcal{L}$. We aim to let the meta-learned policy quickly adapt to this new tool in a fixed number of interactions with the tool in order to maximize the expected cumulative reward. This is a challenging task since the new tool can be quite different in terms of visual appearance and affordances as compared to previously seen tools in meta-training.

# 4 Approach

The key idea behind our approach is to collect and embed language information of the environment into meta-learning, allowing the policy to adapt faster and better to unseen environments.

## 4.1 Collecting Language Information Using Pre-Trained Large Language Models

A common use of language in robotics is to use it to provide an instruction to the robot (*e.g.,* "*pick up the green block on the table*"). Such instructions are typically specified by humans manually through crowd-sourcing, which can be labor intensive. In our setting, we consider language as additional information about the environment (*e.g.,* "*the hammer has a long handle and large head at the top*"). The language here is not used to describe the goal (*e.g.,* what to do), but to provide information about properties of the environment (*e.g.,* tool shape). This makes the text here *task-agnostic*, forcing the agent to learn generalizable policies. To obtain a diverse set of language descriptions, we are inspired by the recent advances in LLMs that are trained with vast amounts of online data and imbued with rich, semantic knowledge of different objects. We propose using LLMs to provide language descriptions of the tools in the form of *question answering*. Specifically, we provide the GPT-3 [41] model with the following template prompt through the OpenAI API:

*"Describe the [feature] of [name] in a detailed and scientific response: "*

where "feature" is selected from one of ["shape", "geometry"] or one of ["common use", "purpose"] and "name" describes the tool (*e.g.,* "*a hammer*", "*a pair of tongs*"). We find that adding "*detailed and scientific*" to the prompt significantly improves the quality of the texts generated. For each tool, we generate 10 different paragraphs of descriptions for each of the four features, and then combine paragraphs in each of the four permutations of the features ("shape" and "common use", etc). Each tool $\tau_i$ is thus paired with a diverse set of 800 language descriptions $L_i$ (see Appendix A1 for more examples). Each description $l_{ij} \sim L_i$ is approximately 2-4 sentences long.

## 4.2 Obtaining Feature Representations from Large Language Models

With the collected language descriptions, we now incorporate them into policy training. One common choice is to train a language module (*e.g.,* long short-term memory (LSTM) [42]) from scratch

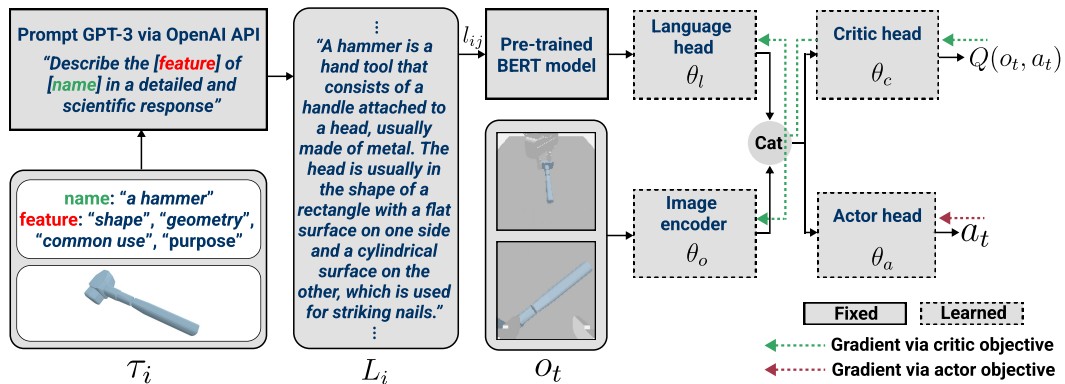

$$\tau_i \qquad\qquad L_i \qquad\qquad O_t$$

Figure 2: **Model Overview.** First, we prompt OpenAI GPT-3 to obtain a diverse set of language descriptions $L_i$ of the tool $\tau_i$. Then for each episode collected, we sample a language description $l_{ij}$ randomly from $L_i$, which is then fed into a pre-trained BERT model to obtain the representation. The language head further distills the language information. We concatenate the representations from the language head and the image encoder, and then the features are shared by the critic head and the actor head.

to embed features of the language input, which can take substantial time and effort to tune. Instead, we use a pre-trained LLM to distill the language descriptions into feature representations. Since LLMs are trained with vast amounts of data, they can better interpret and generalize to the diverse set of long descriptions. Since the GPT-3 model is not publicly available, we opt for the Google BERT-Base [43] model on HuggingFace, which has 110.1M parameters and outputs a 768-dimensional vector representation for each description input. T-SNE analysis shown in Fig. 3 demonstrates that, without any fine-tuning, the model already captures semantic meanings of the descriptions among tools (*e.g.,* hammer and mallet are close to each other).

Fig. 2 shows the overall neural network architecture. We first prompt GPT-3 to obtain a set $L_i$ of text descriptions for tool $\tau_i$ via the procedure in Sec. 4.1. During meta-training, we randomly sample $l_{ij}$ from $L_i$ for each episode to ensure that the policy sees a diverse set of descriptions. We then freeze the BERT model during policy training. The output from BERT is fed into a single fully-connected layer with ReLU (language head, $\theta_l$). The image observations (possibly from two camera angles–one from overhead and one from the wrist) are passed through convolutional layers (image encoder, $\theta_o$), whose output is then concatenated with that from the language head. The actor head ($\theta_a$) and critic head ($\theta_c$) then output the action $a_t$ and the corresponding value for the Q function $Q(o_t, a_t)$. See Appendix A3 for more details of the neural network setup.

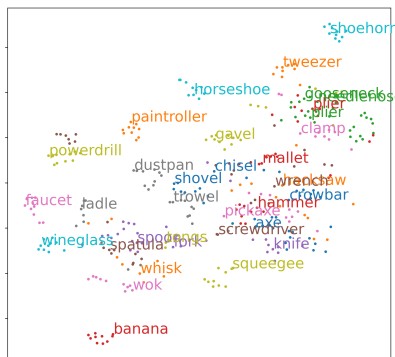

Figure 3: T-SNE results of BERT output for multiple language descriptions of each tool.

### 4.3 Meta-training and Testing Language-Conditioned Manipulation Policies

We hypothesize that additional language information of the tools promotes generalization. However, zero-shot transfer to unseen tools can be difficult given the distinct geometries and affordances. Thus we perform meta-training for explicitly training the policy to adapt to distinct tools within the same task. Our algorithm (shown in Algorithm 1) is based on Reptile [5], a simple first-order gradient-based meta-learning algorithm, but with an additional sampling strategy to prevent overfitting. At each iteration of the meta-training, one tool $\tau$ is sampled from the training set $\mathcal{T}$. At the base level (line 5), we run the current policy with the language description $l_{ij}$ sampled from $L_i$. We then add the collected experiences in a base replay buffer $\beta_{\text{base}}$, and perform $B$ iterations of off-policy updates using Soft Actor Critic (SAC) [44] in order to obtain the final policy parameters $\boldsymbol{\theta}' = [\theta_l', \theta_o', \theta_a', \theta_c']$. Then at the meta level (line 12), the network is updated with a gradient step towards $\theta'$:

$$\boldsymbol{\theta}_{\text{new}} \leftarrow \boldsymbol{\theta} + \alpha(\boldsymbol{\theta}' - \boldsymbol{\theta}), \tag{1}$$

where $\alpha$ is the meta-learning rate and $\boldsymbol{\theta}$ is the collection of the old policy parameters $\theta_l, \theta_o, \theta_a, \theta_c$. We also highlight the following remarks including differences to the Reptile algorithm:

- To reduce variance, the actor and critic share the parameters of language head $\theta_l$ and the image encoder $\theta_o$, and both modules are updated only with the critic objective.

- To prevent overfitting to language descriptions of one tool during training, experiences collected from all tools are saved in a meta replay buffer $\beta_{\text{meta}}$ (line 14), and $\beta_{\text{base}}$ for each tool is initialized with random samples from $\beta_{\text{meta}}$ (line 3). During policy update at base training, 30% of the experiences are sampled from $\beta_{\text{meta}}$. During test time, we do not use any experiences from $\beta_{\text{meta}}$. We demonstrate the effectiveness in Sec. 6.

- In applications of Reptile in supervised learning, the meta update is often averaged over $N$ sampled environments: $\theta_{\text{new}} \leftarrow \theta + \alpha \sum_i^N (\theta_i' - \theta)$. However, we find that $N = 1$ trains faster and also matches the test objective of adapting to a single tool. $N > 1$ does not offer better performance at test time. See Appendix A6 for sensitivity analysis.

- Performing only a single meta update after $B = 5$ iterations of base update can be inefficient. Instead of using a large learning rate $\alpha$ which causes unstable training, we perform $M = 2$ meta updates for each adaptation to one tool, but collect experiences to $\beta_{\text{base}}$ only at the first update.

Algorithm 2 shows the procedure of adaption at test time. First, the model is provided with a target test tool $\tau_\nu$ and a set of language descriptions. With the adaptation budget $B_\nu$, we run the policy with the language description $l_j$ sampled from $L_\nu$. The collected experiences are stored in the buffer $\beta_{\text{base}}$ and used to update the policy parameters.

---

**Algorithm 1** ATLA: Meta-training, $N = 1$

---

**Require:** $\mathcal{T} = \{\tau_i\}_{i=1}^K$: training set of tools; $\{L_i\}_{i=1}^K$: sets of language descriptions; $\theta_l, \theta_o, \theta_a, \theta_c$: policy modules; $\beta_{\text{meta}}$: meta replay buffer.
1: **while** meta-training **do**
2:     Sample $\tau$ from $T$   # meta level
3:     Reset $\beta_{\text{base}}$ with samples from $\beta_{\text{meta}}$
4:     **for** $m = 1$ to $M$ **do**
5:         **for** $b = 1$ to $B$ **do**   # base level
6:             **if** $m = 1$ **then**
7:                 Collect episodes each with $l_{ij} \sim L_i$; add to $\beta_{\text{base}}$
8:             **end if**
9:             Sample from $\beta_{\text{base}}$ and update $\theta_o, \theta_l, \theta_c$ with the critic objective
10:            Sample from $\beta_{\text{base}}$ and update $\theta_a$ with the actor objective
11:         **end for**
12:         Meta update $\theta_o, \theta_l, \theta_c, \theta_a$ with Eq. 1
13:     **end for**
14:     Add $\beta_{\text{base}}$ to $\beta_{\text{meta}}$
15: **end while**

---

**Algorithm 2** Adaption at test time

---

**Require:** $\tau_\nu$: test tool; $L_\nu$: set of language descriptions; $\theta_l, \theta_o, \theta_a, \theta_c$: policy modules; $\beta_{\text{base}} \leftarrow \emptyset$: base replay buffer
1: **for** $b = 1$ to $B_\nu$ **do**
2:     Collect episodes each with $l_j \sim L_\nu$; add to $\beta_{\text{base}}$
3:     Sample from $\beta_{\text{base}}$ and update $\theta_o, \theta_l, \theta_c$ with the critic objective
4:     Sample from $\beta_{\text{base}}$ and update $\theta_a$ with the actor objective
5: **end for**

---

## 5 Experiment Setup

Through different tool manipulation tasks in simulation, we aim to investigate the following questions: (1) Does language information help achieve better adaptation to new tools? (2) Does meta-learning improve adaptation to new tools? (3) How does the choice of pre-trained LLMs affect policy training? (4) Does language information help the policy utilize tools' affordances effectively?

**Tasks.** Four different tool manipulation tasks are implemented (see bottom of Fig. 1): (1) pushing: pushing the tool to a fixed location on the table; (2) lifting: reaching and lifting the tool up from the table to some target height; (3) sweeping: using the tool to sweep a cylinder to a fixed location on the table; (4) hammering: using the tool to hammer a peg further into a hole in a block. Solving these tasks benefits from an understanding of the geometric affordances of the tools such as the grasp location. See App. A2 for more details of the task setup including the reward functions.

**Robot.** We build custom simulation environments with a 7-DOF Franka Panda arm in the PyBullet simulator [45]. We use RGB cameras with $128 \times 128$ image outputs, placed at different off-arm locations and at the arm wrist depending on the needs of the tasks. For all tasks, we use 4-DOF cartesian velocity (3D translation and yaw) as the action output from the policy. The arm joints are then commanded with a jacobian-based velocity controller at 5Hz. The policy does not command the gripper; instead, we use the heuristic that once the gripper is below some height, the gripper closes to grasp the object. If the grasp fails, the gripper re-opens if it rises above the threshold.

**Tools.** We collect a total of 36 objects (See App. A1 for the full list) from open sources. Most of the objects are common tools such as a hammer and an axe. Some of them are less used as tools but have distinct geometry and affordances, such as a banana whose inner curvature may help push other objects. We split the objects into a training set of 27 and a test set of 9 — we try to separate objects with similar geometry or affordances (*e.g.,* hammer and mallet) into different sets.

**Baselines.** We compare ATLA (ours) with the following (Fig. 4): **(a)** *AT-TinyLA* (ours): ATLA with a smaller BERT encoder (BERT-Tiny [43] with 4.4 million parameters and 128-dimensional output). **(b)** *AT*: ATLA without language information. **(c)** *AT-XL*: ATLA without language information but larger networks for $\theta_a$ and $\theta_c$ (matching the number of parameters of ATLA). **(d)** *SAC-LA*: vanilla multi-environment training with SAC and language information but without meta-training objective. **(e)** *SAC*: SAC-LA but without language information. SAC-LA and SAC training follow Algorithm 3 in Appendix A4; they do not perform inner adaptation ($B = 0$) at training time, and the gradient update is averaged over experiences sampled from any environment ($N = \infty$). All meta learning baselines use $B = 5$ and $N = 1$. See Appendix A6 for sensitivity analyses on $N$ and $B$.

**Metric.** For all experiments, we save the model checkpoint with the highest running-average reward on the training dataset. After training, for each test tool we load the checkpoint and run a fixed number of iterations of adaptation. In Fig. 4, we report the highest reward at adaptation, averaged over 3 seeds for each test tool. See Appendix A6 for reward in numbers for each tool and task.

## 6 Results

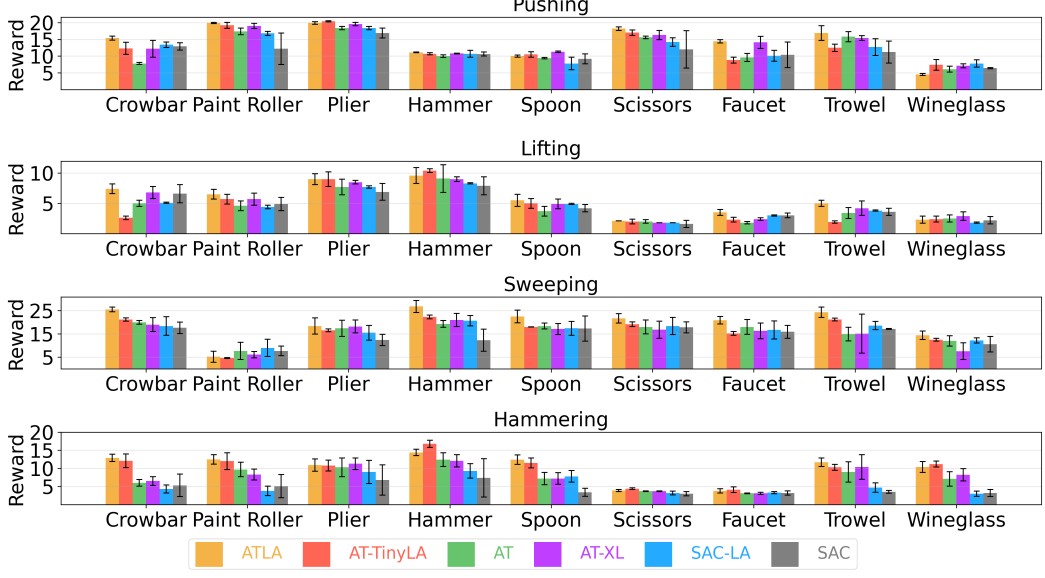

Figure 4: Post-adaptation reward in mean and standard deviation over 3 seeds across 4 tasks and 9 test tools.

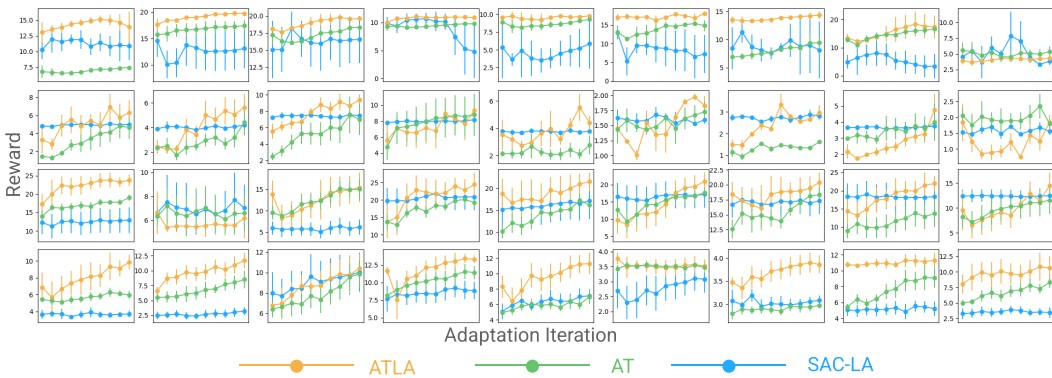

Figure 5: Curves of adaptation iteration vs. reward of the test tools (columns) in the four tasks (rows).

**Q1: Does language information help achieve better adaptation to new tools?** Fig. 4 shows that ATLA and AT-TinyLA perform better than AT and AT-XL among most tools in the 4 tasks. The differences are more significant in sweeping and hammering, which are more difficult and language information can better inform the affordances of the tools. Among the 9 tools, ATLA and AT-TinyLA always perform better with crowbar, plier, hammer, scissors, faucet, and trowel. ATLA and AT-TinyLA do not perform better mostly when low reward is achieved for all baselines for that tool, such as wineglass in pushing and lifting and paint roller in sweeping. Fig. 5 also shows that language helps faster learning in ATLA compared to AT in most cases, with the agent achieving higher rewards with fewer episodes of adaptation.

We also find that different tools learn better for different tasks. For example, hammer is better in sweeping than plier probably due to its long bar, but worse in pushing also due to the small inertia along the long bar causing instability during pushing. As ATLA performs better among most tools, language information can provide useful information about the tool affordances in different tasks.

**Q2: Does meta-learning improve adaptation to new tools?** Fig. 4 shows that with or without language information (ATLA / AT-TinyLA vs. SAC-LA, or AT / AT-XL vs. SAC), meta-learning improves final performance after adaptation. Without meta-learning, SAC-LA shows smaller improvement over SAC (*e.g.,* plier, hammer, scissors, and trowel). This demonstrates that language information particularly helps when combined with meta-learning.

Fig. 5 also compares the adaptation curves between ATLA and SAC-LA: those of SAC-LA tend to stagnate or fluctuate while those of ATLA tend to rise steadily. This indicates meta-learning trains the policy to better adapt to new tools after training. In Appendix A6 we also perform sensitivity analysis on the number of inner adaptation during meta-training, $B$, which highlights the effectiveness of performing a few steps of inner adaptation.

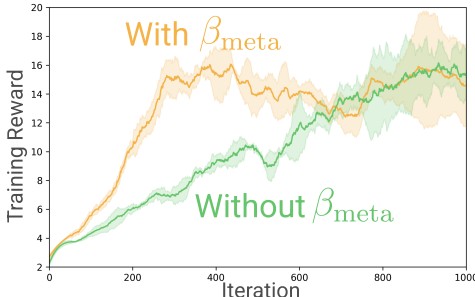

Figure 6: Using meta replay buffer also accelerates training.

**Ablation: meta replay buffer.** We investigate the effect of re-using experiences from other tools (saved in $\beta_{\text{meta}}$ during adaptation). For this, we run ATLA without $\beta_{\text{meta}}$ on the pushing task. Across the 9 test tools, the post-adaptation reward is mostly lower ($-20\%, -10\%, -13\%, -18\%, +5\%, -25\%, -18\%, -18\%, -5\%$) compared to ATLA with $\beta_{\text{meta}}$. Note that the effect is more prominent when the reward difference between ATLA and AT is larger (*e.g.,* 20% with crowbar and 25% with scissors), indicating that language information is more effective if $\beta_{\text{meta}}$ is applied. We also find using $\beta_{\text{meta}}$ accelerates the meta-learning process (Fig. 6) — demonstrating that sharing experiences among tools makes training more efficient.

**Q3: How does the choice of pre-trained LLMs affect policy training?** Fig. 4 shows that ATLA usually attains higher post-adaptation reward than AT-TinyLA, which uses a smaller pre-trained BERT model, indicating that the policies benefit from the richer representation of the language descriptions that the bigger BERT model offers.

**Q4: Does language information help the policy utilize tools' affordances effectively?** The results above have shown that language descriptions of the geometric features and common use of the tools help policies adapt to new tools for a given task. Fig. 7(a) visualizes the effect in the example of using a crowbar for sweeping. Language descriptions of a crowbar often contain phrases including "long and thin bar", "curved", "hook", "used to leverage", and "used to pry open things". With the descriptions, ATLA (orange curve in Fig. 7(a)) enables the policy to adapt quickly to this tool unseen during meta-training — the policy learns to use the curved hook to better steer the cylinder towards the target. As a comparison, we replace the descriptions with only the sentence "A crowbar is a long and thin bar," and the policy (green curve in Fig. 7(a)) does not adapt as well.

One common feature among tools is the handle. Language descriptions of a trowel includes phrases like "flat, triangular blade", "handle to be grasped", and "used for scooping". While ATLA learns to grasp at the handle (Fig. 7(b) top), when we remove "handle" from all the descriptions, the robot fails to grip firmly on the handle and loses the grip eventually (Fig. 7(b) bottom).

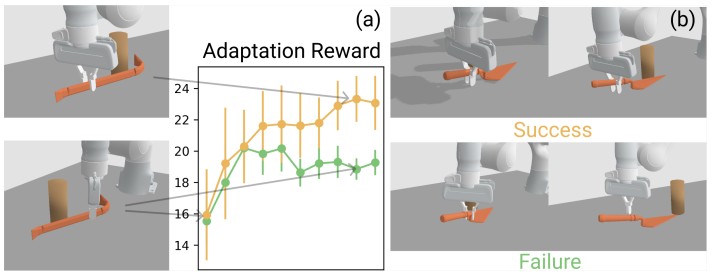

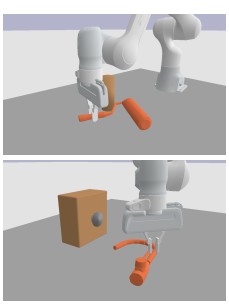

Figure 7: With language information, ATLA is able to adapt the policy to utilize the affordances of the tools – (a) curved hook on a crowbar; (b) handle on a trowel.

Figure 8: Policies may fail to correctly utilize affordances.

**Limitations.** In some cases, we observe that the policies can still fail to correctly utilize the tool affordances with language information. In the sweeping task, the paint roller is the only tool that ATLA fails to perform the best with. Fig. 8 (top) shows the grasp learned by the policy. It failed to use the bigger opening between the roller and the handle on the other side to sweep the cylinder. Fig. 8 (bottom) shows that in the hammering task, the policy fails to use the faucet, specifically, its relatively flat head to push towards the nail. However, these affordances can be sensitive to the initial pose of the tool and can be difficult to explore. This also highlights one of the limitations of our work. We use a relatively simple task policy setup for tool manipulation tasks, which is directly mapping image inputs to Cartesian velocity commands. This creates challenges in exploration and learning the skills even with language information. One remedy is to combine with approaches like keypoint-based methods [7] that inject additional domain knowledge into the policy. In addition, we use a relatively small dataset of tools, which may limit the potential of using language information. It would be particularly interesting to model revolute joints of the tools and perform more complex tasks such as picking up objects with a pair of tongs. Furthermore, our current evaluation does not consider real robot experiments, which is *not* the focus of this work.

## 7 Conclusion

In this work, we investigate using large language models (LLMs) to accelerate adaptation of policies to new tools in tool manipulation tasks. We use LLMs to both (1) generate diverse language descriptions of the tool geometry and common use, and (2) obtain vector representations of the descriptions. We then propose language-conditioned meta-learning that trains policies to quickly adapt to new tools. The results demonstrate that combining language information and meta-learning significantly improves the performance when adapting to unseen tools.

**Acknowledgments**

The authors were partially supported by the Toyota Research Institute (TRI), the NSF CAREER Award [#2044149], the Office of Naval Research [N00014-21-1-2803, N00014-18-1-2873], and the School of Engineering and Applied Science at Princeton University through the generosity of William Addy '82. This article solely reflects the opinions and conclusions of its authors and not NSF, ONR, Princeton SEAS, TRI or any other Toyota entity.

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

# Appendix

## A1 Tool Information

Table A1 shows additional information about the tools used in the paper, including a sample view of the object mesh, sample language descriptions, and the train-test split.

| Tool | Sample view | Sample Language description | Used for testing |
|---|---|---|---|
| axe | | An axe typically has a long, cylindrical handle with a flared end. The head of the axe typically has a slightly curved blade. An axe is often used for chopping wood. | No |
| chisel | | Chisel can make clean, precise cuts using the beleveled edge. A chisel is a hand tool with a blade attached to a handle. | No |
| crowbar | | A crowbar is used as a lever to pry things open. One end of a crowbar is usually curved or hooked so that it can be jammed under an object to apply leverage. | Yes |
| shovel | | A shovel has a long, cylindrical handle and a scoop-shaped blade. The shovel typically has a curved part for digging into and scooping up materials. | No |
| hacksaw | | A hacksaw is a hand saw with a thin blade attached to a handle, used for cutting various such as metal, plastic, or wood. A hacksaw is a saw with a thin, toothed blade on a rigid frame, used for cutting wood or metal. | No |
| paintroller | | One common use of a paint roller is to apply paint evenly to a surface such as walls or ceilings. A paint roller consists of a long, cylindrical body with a handle on one end. | Yes |
| tweezer | | A tweezer is a hand-held tool with two arms that meet at a point. A tweezer is a small hand-held tool with two pointed jaws that are used to pick up small objects or to remove unwanted hair or debris from the body. | No |
| whisk | | A whisk typically has a long, thin handle with a series of loops at the end. The loops are usually made of metal and are arranged in a spiral pattern. A whisk is a common kitchen utensil that is used to mix ingredients together or to incorporate air into a mixture. | No |
| needlenose | | A needlenose plier has a long, tapered nose with a small jaw, and is used for gripping and bending wire. Needlenose pliers are a type of plier that has a long, slender nose and is used for gripping small objects and for working in tight spaces. | NO |
| plier | | A plier is a hand tool used for gripping objects. It consists of a pair of metal jaws with teeth that open and close when the handles are moved. Plier typically has a long, narrow neck and a tapered head that becomes progressively thinner as it extends from the neck to the tip. | Yes |
| gooseneck | | A gooseneck plier is a type of plier that has a long, narrow neck and a slightly curved head. The neck allows the plier to reach into tight spaces, and the curved head provides extra leverage. A gooseneck plier is commonly used to grip and bend small objects. | No |
| plier-open | | The shape of a plier is typically long and skinny with a grip at the end. Plier is a hand tool used for various purposes such as gripping, bending and cutting. | No |

| | | | |
|---|---|---|---|
| mallet |  | A mallet is a tool that is used to strike another object. A mallet is a type of hammer that usually has a large head and a long handle. | No |
| hammer |  | The purpose of a hammer is to strike or hit another object. A hammer typically has a long, cylindrical handle and a heavy head. | Yes |
| banana |  | The shape of a banana is generally long and curved, with a thin skin and fleshy inside. A banana is a curved, yellow fruit with a thick peel. | No |
| fork |  | A fork is long and thin, with three tines (prongs) at the end. A fork is a utensil that consists of a handle with several narrow tines on one end. The tines are used for piercing food and then lifting it to the mouth. | No |
| spoon |  | The purpose of a spoon is to transfer a liquid or semi-solid food from a container to the mouth. A typical spoon consists of a bowl-shaped container with a handle extending from one side. The bowl is generally oval or round, and the handle generally tapers towards the end. | Yes |
| knife |  | A knife typically has a sharp, narrow blade with a pointed tip. A knife is a common kitchen utensil used for cutting and slicing food. | No |
| spatula |  | A spatula is a kitchen utensil that is used to turn or lift food that is being cooked. It has a flat, usually slightly convex, blade that is attached to a handle. A spatula is commonly used to mix, spread, and flip food items. | No |
| scissors |  | A pair of scissors is a cutting tool that consists of two metal blades that are connected at a pivot point. A pair of scissors typically has two blades that are joined at a pivot point. | Yes |
| wrench |  | A wrench is a tool that is used to apply torque to an object in order to loosen or tighten it. A wrench is typically long and slender with a small, metal handle. | No |
| screwdriver |  | The geometry of a screwdriver can be described as a cylindrical shape with a pointed end. A screwdriver is a tool that is used to insert and remove screws. | No |
| clamp |  | A clamp is a mechanical device that is used to temporarily hold two or more objects together. The geometry of a clamp is typically that of a rectangular or U-shaped object with two handles. | No |
| wok |  | The shape of a wok is a deep, round bowl with sloping sides. A wok is a concave-shaped cooking utensil that is most commonly used in Chinese cuisine. | No |
| pickaxe |  | A pickaxe is used to break up rocks and other materials. A pickaxe is a tool that has a handle attached to a head. | No |
| faucet |  | A faucet is typically a small, thin, spout-like fixture that protrudes from a wall or sink. A faucet is a valve used to release water from a plumbing fixture, such as a sink or bathtub. | Yes |
| dustpan |  | A dustpan is a tool used for sweeping up dust and small debris from floors and other surfaces. It consists of a small, shallow pan with a handle attached to one side. A dustpan is a concave scoop with a flat bottom and flared sides. | No |

| | | | |
|---|---|---|---|
| trowel | | A trowel is generally a small hand tool with a pointed, scoop-shaped blade on one end and a flat surface on the other. A trowel is a small, hand-held gardening tool with a curved, pointed blade that is used for digging, planting, and transferring small amounts of soil or other materials. | Yes |
| ladle | | A ladle is a tool used to transfer liquids from one container to another. A ladle typically has a long, curved handle and a large, deep, spoon-like bowl. | No |
| tongs | | A pair of tongs has a thin, curved metal shaft with two flat metal paddles at the end. A pair of tongs is a device used to grip and hold objects. | No |
| gavel | | A gavel is a small hammer that is used to strike a sound block, typically made of wood. A gavel is a mallet used to strike a block of wood, typically used by a presiding officer or auctioneer to maintain order or to signal the start and end of an auction. | No |
| squeegee | | The purpose of a squeegee is to remove water or other liquid from a surface. A squeegee is a rod-shaped tool with a flat, blunt edge, and a small handle. | No |
| powerdrill | | A powerdrill is typically cylindrical in shape, with a handle attached to one side and a chuck on the other side for holding drill bits. A power drill is a tool that is used to create holes in various materials, or to fasten screws or bolts. | No |
| wineglass | | A wineglass is a glass with a small bowl and a long stem. They are used to serve wine and are often used in restaurants. A wineglass is typically shaped with a long, thin stem and a bowl that is larger at the bottom than the top. | Yes |
| shoehorn | | A shoehorn is a curved, rod-shaped object used to assist in putting on shoes. A shoehorn is a curved or stepped tool designed to help slide a shoe onto the foot. | No |
| horseshoe | | A horseshoe is a U-shaped metal bar that is nailed to the hooves of a horse. A horseshoe is typically U-shaped, with two large curves and two smaller curves at either end. | No |

Table A1: Sample views, sample language descriptions, and the train-test split of the 36 tools considered in the paper.

Fig. 3 shows the t-SNE analysis of the BERT embeddings of all the tools. First we use PCA to project the 768-dimensional embeddings to 50-dimensional, and then perform t-SNE to project them to 2-dimensional for visualization.

## A2 Task Information

Table. A2 shows the episode length, reward function, and action space of the tasks. We find the policy can explore well in pushing and lifting tasks with relatively simple reward functions; in sweeping and hammering task, we tune the reward function carefully to guide the arm towards the cylinder/nail. Fig. A1 shows the camera observations for the four tasks. We use a single view for the pushing task as it is sufficient for the task, and dual views for other tasks. A wrist view is used in the lifting task. Fig. A2 visualizes the workspace of the tasks including the initial position of the tools and the target.

For the hammering task, we set the lateral and torsional friction coefficient of the nail to be high (1 and 0.1) in the simulator. We also make the gripper fingers longer to prevent the gripper hitting the block when attempting to hammer the nail.

Please see the included video for more visualization of the tasks.

| Task | Episode length | Reward function | Action space |
|---|---|---|---|
| Pushing | 25 | $\max(0, 1 - \text{distance-tool-target})$ | $[-0.05, 0.15]m/s$ in $x$ $[-0.1, 0.1]m/s$ in $y$ $[-\pi/4, \pi/4]\text{rad}/s$ in yaw |
| Lifting | 25 | $0.1 * \max(0, 1 - \text{distance-EE-tool})+$ $0.5 * \max(0, 1 - \text{distance-tool-target})$ | $[-0.1, 0.1]m/s$ in $x$, $y$, and $z$ $[-\pi/4, \pi/4]\text{rad}/s$ in yaw |
| Sweeping | 40 | $0.1 * \max(0, 1 - \text{distance-EE-tool})+$ $0.1 * \max(0, 1 - \text{distance-tool-cylinder})+$ $0.5 * \max(0, 1 - \text{distance-cylinder-target})$ | $[-0.2, 0.2]m/s$ in $x$, $y$, and $z$ $[-\pi/4, \pi/4]\text{rad}/s$ in yaw |
| Hammering | 40 | $0.1 * \max(0, 1 - \text{distance-EE-tool})+$ $0.1 * \max(0, 1 - \text{distance-tool-nail})+$ $0.5 * \max(0, 1 - \text{distance-nail-hole\_end})$ | $[-0.2, 0.2]m/s$ in $x$, $y$, and $z$ $[-\pi/4, \pi/4]\text{rad}/s$ in yaw |

Table A2: Episode length, reward function, and action space for the four tasks. Distance-{A}-{B} denotes distance from A to B, normalized by the initial distance. EE denotes end-effector of the arm. See Fig. A2 for visualization of the task space and target.

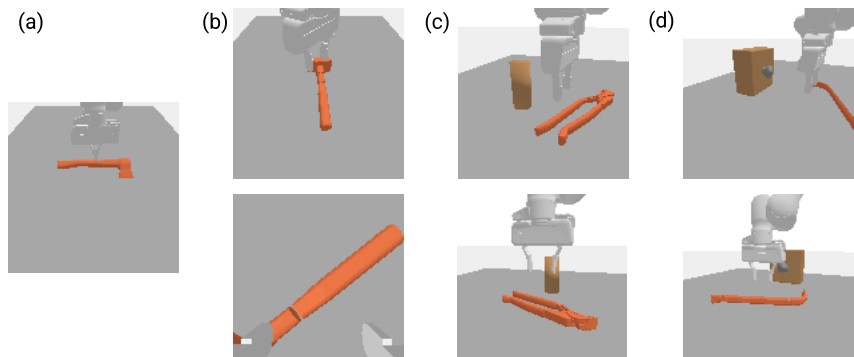

Figure A1: Camera observations of the tasks: (a) pushing (single view only); (b) lifting (including a wrist view); (c) sweeping; (d) hammering.

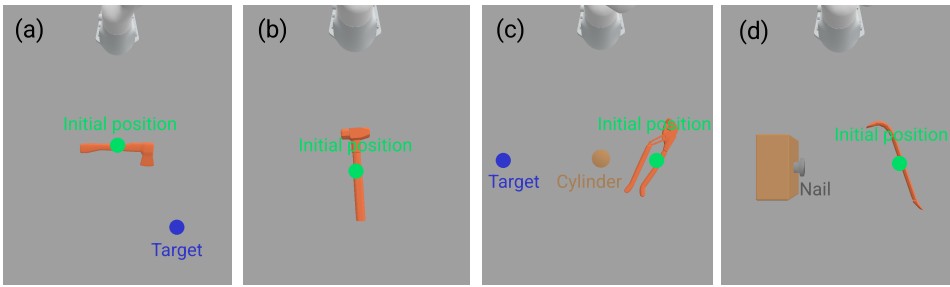

Figure A2: Top-down visualization of the workspace of the tasks: (a) pushing; (b) lifting; (c) sweeping; (d) hammering.

## A3   Model Architecture

For all policies, the image encoder $\pi_o$ contains three convolutional layers for either view of the image input; the three layers have kernel size $[7 \times 7, 5 \times 5, 3 \times 3]$, stride size $[4, 3, 2]$, no padding, and channel size $[4, 8, 16]$. The language head $\pi_l$ contains a single fully-connected layers with 128-dimensional output. Both the actor head $\pi_a$ and critic head $\pi_c$ have two hidden layers of size 128 (except for AT-XL with hidden size 256). All convolutional layers and fully-connected layers are followed with a ReLU activation. The first layer in $\pi_a$ and $\pi_c$ are additionally normalized by Layernorm [46].

## A4 Multi-environment Training Algorithm

Algorithm 3 below shows the procedures of the SAC and SAC-LA baselines. Multi-environment training does not involve any inner adaptation ($B = 0$). In the context of multi-environment training, $N$ means the number of environments where, experiences that each gradient update uses, belong. $N = \infty$ means experiences can come from any environment.

---

**Algorithm 3** SAC, SAC-LA

---

**Require:** $\mathcal{T} = \{\tau_i\}_{i=1}^K$: training set of tools; $\{L_i\}_{i=1}^K$: sets of language descriptions (for SAC-LA only); $\theta_l$ (for SAC-LA only), $\theta_o, \theta_a, \theta_c$: policy modules; $\beta$: replay buffer; $N$
1: **while** training **do**
2:     Sample $\tau_i$ from $T$; reset environment with $\tau_i$
3:     Collect episode (with $l_{ij} \sim L_i$ for SAC-LA); add to $\beta$
4:     **if** $N = \infty$ **then**
5:         Sample experiences from $\beta$ that are from any environment
6:     **else**
7:         Sample experiences from $\beta$ that are from $N$ different environments
8:     **end if**
9:     Update $\theta_o, \theta_l$ (only for SAC-LA), $\theta_c, \theta_a$ with sampled experiences
10: **end while**

---

## A5 Training Hyper-parameters

The hyper-parameters used for meta-learning (shared among AT-LA, AT-TinyLA, AT, AT-XL) and multi-environment learning (shared between SAC-LA, SAC) are outlined in Table A3. We ensure meta-learning and multi-environment learning sample the same amount of transitions from environments.

| Setting | Meta-learning | Multi-environment learning |
|---|---|---|
| # training steps | 1000 (iteration) | 2.5e6 (pushing/lifting) 4e6 (sweeping/hammering) |
| Meta replay buffer size | 30000 | — |
| Base replay buffer size | $\infty$ | 100000 |
| Replay ratio | | 16 |
| $N$ | 1 | $\infty$ |
| $M$ | 2 | — |
| $B$ | 5 | 0 |
| $B_\nu$ | | 10 |
| **Optimization** | | |
| Optimizer | | Adam |
| Batch size | | 128 |
| Discount factor | | 0.99 |
| SAC entropy coefficient | | 0.01 |
| SAC actor update period | | 1 |
| Base learning rate | | 3e-4 |
| Meta learning rate | 1e-3 | — |
| **Hardware Resource** | | |
| # CPU threads | | 20 |
| GPU | | Nvidia RTX 2080Ti |
| # hours for runtime | 6 (pushing/lifting), 10 (sweeping), 16 (hammering) | |

Table A3: Hyper-parameters used for meta-learning and multi-environment learning.

## A6 Results and Additional Studies

Table A4 below shows the results in Fig. 4 in numbers.

| Method | Crowbar | Paint Roller | Plier | Hammer | Spoon | Scissors | Faucet | Trowel | Wineglass |
|---|---|---|---|---|---|---|---|---|---|
| | **Pushing** | | | | | | | | |
| ATLA | **15.4 ± 0.6** | **19.9 ± 0.2** | 19.9 ± 0.4 | **11.1 ± 0.1** | 10.0 ± 0.3 | **18.2 ± 0.5** | **14.4 ± 0.5** | **16.9 ± 2.2** | 4.5 ± 0.3 |
| AT-TinyLA | 12.3 ± 1.8 | 19.2 ± 0.9 | **20.4 ± 0.2** | 10.7 ± 0.3 | 10.5 ± 0.8 | 17.0 ± 0.8 | 8.8 ± 0.9 | 12.5 ± 1.1 | 7.4 ± 1.6 |
| AT | 7.8 ± 0.3 | 17.4 ± 1.0 | 18.4 ± 0.5 | 10.0 ± 0.4 | 9.4 ± 0.2 | 15.6 ± 0.4 | 9.6 ± 1.2 | 15.8 ± 1.5 | 6.1 ± 0.9 |
| AT-XL | 12.2 ± 2.5 | 19.0 ± 0.8 | 19.6 ± 0.5 | 10.8 ± 0.1 | **11.3 ± 0.2** | 16.3 ± 1.4 | 14.1 ± 1.8 | 15.4 ± 0.7 | 7.1 ± 0.6 |
| SAC-LA | 13.4 ± 0.8 | 16.8 ± 0.6 | 18.4 ± 0.5 | 10.7 ± 1.0 | 7.8 ± 1.9 | 14.2 ± 1.3 | 10.1 ± 1.6 | 12.7 ± 2.5 | 7.8 ± 1.1 |
| SAC-LA-N=1 | 13.8 ± 0.9 | 15.9 ± 0.3 | 17.8 ± 0.7 | 10.5 ± 1.1 | 8.2 ± 2.0 | 13.2 ± 1.0 | 10.3 ± 1.8 | 14.2 ± 3.0 | **7.9 ± 1.2** |
| SAC | 12.9 ± 1.1 | 12.2 ± 4.7 | 16.9 ± 1.5 | 10.6 ± 0.6 | 9.2 ± 1.5 | 12.0 ± 5.6 | 10.4 ± 3.8 | 11.2 ± 3.3 | 6.4 ± 0.2 |
| | **Lifting** | | | | | | | | |
| ATLA | **7.4 ± 0.8** | **6.5 ± 0.8** | **9.0 ± 0.9** | 9.6 ± 1.3 | **5.5 ± 1.0** | **2.1 ± 0.0** | **3.5 ± 0.5** | **5.0 ± 0.5** | 2.3 ± 0.6 |
| AT-TinyLA | 2.6 ± 0.3 | 5.7 ± 0.8 | 9.0 ± 1.2 | **10.4 ± 0.3** | 5.0 ± 0.8 | 2.0 ± 0.4 | 2.3 ± 0.4 | 1.9 ± 0.2 | 2.4 ± 0.5 |
| AT | 5.0 ± 0.5 | 4.6 ± 0.8 | 7.7 ± 1.3 | 9.1 ± 2.3 | 3.7 ± 0.8 | 2.0 ± 0.3 | 1.8 ± 0.2 | 3.4 ± 0.9 | **2.5 ± 0.6** |
| AT-XL | 6.8 ± 1.0 | 5.7 ± 1.0 | 8.5 ± 0.3 | 9.0 ± 0.4 | 4.9 ± 0.8 | 1.8 ± 0.0 | 2.4 ± 0.2 | 4.2 ± 1.2 | 2.9 ± 0.7 |
| SAC-LA | 5.1 ± 0.1 | 4.4 ± 0.3 | 7.7 ± 0.2 | 8.3 ± 0.1 | 4.9 ± 0.1 | 1.8 ± 0.0 | 3.0 ± 0.1 | 3.8 ± 0.1 | 1.8 ± 0.1 |
| SAC-LA-N=1 | 5.5 ± 0.3 | 4.2 ± 0.3 | 8.0 ± 0.5 | 8.1 ± 0.2 | 4.5 ± 0.1 | 1.8 ± 0.0 | 3.2 ± 0.1 | 3.6 ± 0.2 | 1.9 ± 0.2 |
| SAC | 6.6 ± 1.5 | 4.9 ± 1.1 | 6.9 ± 1.4 | 7.9 ± 1.5 | 4.2 ± 0.6 | 1.6 ± 0.6 | 3.0 ± 0.4 | 3.6 ± 0.6 | 2.2 ± 0.6 |
| | **Sweeping** | | | | | | | | |
| ATLA | **25.5 ± 1.0** | 5.2 ± 2.4 | **18.4 ± 3.5** | **26.8 ± 2.6** | **22.5 ± 2.7** | **21.7 ± 2.0** | **20.9 ± 1.6** | **24.3 ± 2.2** | **14.4 ± 1.8** |
| AT-TinyLA | 21.2 ± 0.7 | 4.6 ± 0.2 | 16.5 ± 0.6 | 22.3 ± 0.8 | 18.0 ± 0.1 | 19.2 ± 1.0 | 15.2 ± 0.8 | 21.2 ± 0.6 | 12.5 ± 0.4 |
| AT | 20.0 ± 0.8 | 7.7 ± 3.7 | 17.4 ± 3.5 | 19.3 ± 1.5 | 18.4 ± 1.3 | 18.0 ± 3.0 | 18.0 ± 3.2 | 14.9 ± 2.9 | 12.0 ± 2.2 |
| AT-XL | 19.0 ± 3.0 | 6.1 ± 1.4 | 18.2 ± 2.8 | 21.0 ± 2.8 | 17.1 ± 2.4 | 16.8 ± 3.7 | 16.3 ± 3.4 | 15.1 ± 8.4 | 7.6 ± 3.6 |
| SAC-LA | 18.4 ± 4.0 | **9.0 ± 3.7** | 15.5 ± 3.2 | 20.7 ± 2.2 | 17.4 ± 3.0 | 18.4 ± 3.7 | 16.7 ± 2.9 | 18.6 ± 1.8 | 12.2 ± 1.1 |
| SAC-LA-N=1 | 16.8 ± 3.2 | 7.5 ± 2.2 | 16.1 ± 3.5 | 21.9 ± 1.7 | 17.1 ± 2.8 | 17.9 ± 4.0 | 17.1 ± 2.6 | 18.1 ± 1.9 | 12.6 ± 1.6 |
| SAC | 17.6 ± 2.5 | 7.7 ± 2.1 | 12.4 ± 2.4 | 12.3 ± 4.7 | 17.3 ± 5.4 | 17.8 ± 2.4 | 15.9 ± 2.8 | 17.1 ± 0.2 | 10.6 ± 3.3 |
| | **Hammering** | | | | | | | | |
| ATLA | **12.9 ± 1.0** | **12.5 ± 1.3** | **10.9 ± 1.7** | 14.4 ± 0.9 | **12.4 ± 1.3** | 3.9 ± 0.3 | 3.8 ± 0.6 | **11.7 ± 1.2** | 10.4 ± 1.5 |
| AT-TinyLA | 12.1 ± 1.9 | 12.0 ± 2.3 | 10.8 ± 1.5 | **16.8 ± 1.0** | 11.5 ± 1.4 | **4.4 ± 0.2** | **4.1 ± 0.8** | 10.3 ± 0.8 | **11.2 ± 0.8** |
| AT | 6.0 ± 0.9 | 9.7 ± 2.0 | 10.3 ± 2.6 | 12.4 ± 1.9 | 7.2 ± 1.7 | 3.7 ± 0.1 | 3.1 ± 0.1 | 9.0 ± 2.8 | 7.1 ± 2.0 |
| AT-XL | 6.5 ± 1.2 | 8.3 ± 1.5 | 11.3 ± 1.6 | 12.1 ± 1.7 | 7.2 ± 1.6 | 3.7 ± 0.1 | 3.1 ± 0.3 | 10.4 ± 3.4 | 8.2 ± 1.7 |
| SAC-LA | 4.3 ± 1.1 | 3.8 ± 1.3 | 9.0 ± 3.2 | 9.3 ± 2.0 | 7.8 ± 1.6 | 3.2 ± 0.5 | 3.3 ± 0.3 | 4.7 ± 1.3 | 3.0 ± 0.7 |
| SAC-LA-N=1 | 5.3 ± 1.5 | 6.9 ± 1.1 | 10.0 ± 2.5 | 11.5 ± 1.5 | 9.8 ± 2.9 | 3.6 ± 0.7 | 3.8 ± 0.2 | 8.9 ± 1.6 | 6.8 ± 1.0 |
| SAC | 5.3 ± 3.1 | 5.1 ± 3.2 | 6.8 ± 4.2 | 7.4 ± 5.3 | 3.4 ± 1.1 | 3.0 ± 0.6 | 3.2 ± 0.6 | 3.5 ± 0.4 | 3.2 ± 1.0 |

Table A4: Post-adaptation reward in mean and standard deviation over 3 seeds across 4 tasks and 9 test tools.

**Effect of $B$ in meta-learning.** The value of $B$ determines the number of inner adaptation for each tool during meta-training. To evaluate the effect of $B$, we perform sensitivity analysis varying $B$ in the sweeping task, and the results are shown in Table A5 (values for $B = 5$ from Table A4). With $B = 1$, the post-adaptation performance is generally worse than $B > 1$, and it is close to the performance of multi-environment training (SAC-LA in Table A4). Higher value of $B$ helps, but there is no significant performance gain with $B > 2$. This highlights the importance of performing a few steps of inner adaptation at training time for better adaptation to new tools.

| B | Crowbar | Paint Roller | Plier | Hammer | Spoon | Scissors | Faucet | Trowel | Wineglass |
|---|---|---|---|---|---|---|---|---|---|
| | **Sweeping** | | | | | | | | |
| 1 | 18.6 ± 1.1 | 7.2 ± 2.5 | 16.1 ± 2.2 | 20.9 ± 1.8 | 19.3 ± 1.8 | 18.2 ± 2.9 | 16.2 ± 0.8 | 19.3 ± 2.2 | 12.7 ± 1.1 |
| 2 | 21.2 ± 0.9 | **7.3 ± 2.0** | 17.2 ± 2.5 | 22.8 ± 2.1 | 22.1 ± 2.1 | **22.8 ± 2.3** | 18.9 ± 1.8 | 21.2 ± 2.8 | 13.8 ± 1.7 |
| 4 | 22.6 ± 0.8 | 5.9 ± 1.8 | 17.6 ± 2.3 | **27.8 ± 2.9** | 20.1 ± 3.0 | 20.2 ± 2.2 | **21.1 ± 1.9** | 23.2 ± 2.9 | **14.6 ± 1.9** |
| 5 | **25.5 ± 1.0** | 5.2 ± 2.4 | **18.4 ± 3.5** | 26.8 ± 2.6 | **22.5 ± 2.7** | 21.7 ± 2.0 | 20.9 ± 1.6 | **24.3 ± 2.2** | 14.4 ± 1.8 |

Table A5: Effect of $B$ in meta-learning. The values are post-adaptation reward in mean and standard deviation over 3 seeds across 9 test tools in the sweeping task.

**Effect of $N$ in meta-learning.** The value of $N$ determines how many environments the meta update gradient is averaged over in Eq. 1. To evaluate the effect of $N$, we perform sensitivity analysis varying $N$ in the hammering task, and the results are shown in Table A6 (values for $N = 1$ are from Table A4). Although typical Reptile-style meta learning in supervised learning uses $N > 1$ [5], the results here do not show the improvement with $N > 1$. $N = 1$ also matches our setup of adapting to a single tool at test time. We also show the meta training reward curves in Fig. A3. The final training rewards are similar for $N = 1, 2, 5$, but $N = 1$ trains slightly faster. Surprisingly we also find $N > 1$ exhibits larger variances in reward at later iterations, which is contrary to the idea that $N > 1$ stabilizes training. Thus we use $N = 1$ in our main experiments.

| | | | | **Hammering** | | | | | |
|---|---|---|---|---|---|---|---|---|---|
| N | Crowbar | Paint Roller | Plier | Hammer | Spoon | Scissors | Faucet | Trowel | Wineglass |
| 1 | **12.9 ± 1.0** | 12.5 ± 1.3 | 10.9 ± 1.7 | **14.4 ± 0.9** | 12.4 ± 1.3 | 3.9 ± 0.3 | **3.8 ± 0.6** | 11.7 ± 1.2 | **10.4 ± 1.5** |
| 2 | 12.2 ± 1.2 | **12.9 ± 1.5** | 9.7 ± 1.2 | 14.2 ± 1.1 | 12.9 ± 1.5 | 4.0 ± 0.2 | 3.6 ± 0.3 | **12.0 ± 1.6** | 10.1 ± 1.8 |
| 5 | 12.5 ± 1.0 | 12.5 ± 1.7 | **11.2 ± 1.3** | 11.5 ± 1.2 | **13.2 ± 1.1** | **4.2 ± 0.5** | 3.7 ± 0.6 | **12.0 ± 1.5** | 9.9 ± 1.3 |

Table A6: Effect of $N$ in meta-learning. The values are post-adaptation reward in mean and standard deviation over 3 seeds across 9 test tools in the hammering task.

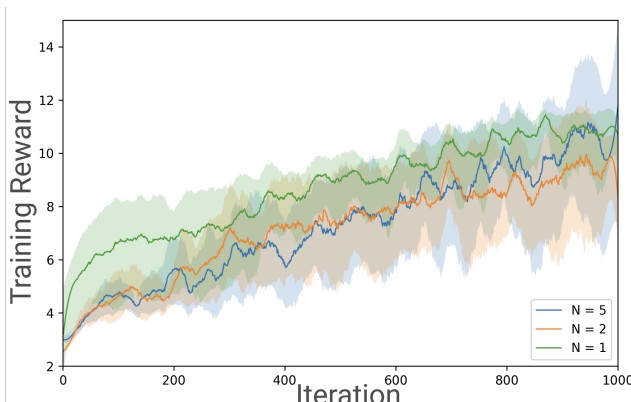

Figure A3: Meta training reward for different values of $N$ in the hammering task running ATLA.

**SAC-LA with $N = 1$.** Typically multi-environment training uses gradient update averaged over a batch of experiences sampled from any environment ($N = \infty$); we perform the same setup in our main experiments. It is also worth investigating $N = 1$, meaning each gradient update uses experiences from only one environment, since we find $N = 1$ works for the meta learning setup as shown above. In Table A4 we show the results for all four tasks (SAC-LA-N=1). In Pushing, Lifting, and Sweeping tasks, the results are similar to SAC-LA with $N = \infty$. However, in the hammering task, the results are improved from SAC-LA across all tools, although they are still worse than meta-learning baselines. It is possible that $N = 1$ mitigates training instability of multi-environment training. We use $N = \infty$ in our main experiments since it is a common setup for multi-environment training.

