# OpenReview forum: "Leveraging Language for Accelerated Learning of Tool Manipulation"
_robot-learning.org/CoRL/2022/Workshop/LangRob — LangRob 2022 Poster_

### Official Review · Reviewer_xbxp · 2022-11-08
**Okey Paper**

**Rating:** 6
**Confidence:** 3

**Review:**

The paper presents a novel method in which textual information about tools is generated with GPT-3, encoded with a pretrained BERT LLM and then a policy is trained with meta-learning. The paper is well written and the approach is sound. Some comments and questions:
1) Recent studies have shown[0] that in order to better capture the semantics of long sentences, LLMs that have been finetuned on sentence similarity show better performance for downstream control tasks. You might want to try one such model instead of BERT. From the experiments is a bit unclear how important the GPT-3 generated sentences are vs, just hand-labeling "this is a hammer to nail stuff" etc.
2) Is there a reason why the visual encoder is trained from scratch? For example, would your approach not benefit from using pretrained VLMs as initialization, such as CLIP features?
3) There are multiple works that use visual-lingual affordances for downstream robot control tasks, such as Cliport[1] or HULC++[2]. Maybe you could briefly mention how they relate to your approach and check if they might serve as a baseline? I ask because currently, the baselines presented are just ablations of the main method.
4) In general, although the idea is nice the results do not seem super convincing, it's hard to see big differences in Figure 4 across the methods. Also as you plot the reward in the y axis, it's difficult to an average over the different tasks. It would be helpful to have normalized rewards between 0 to 1 for this.
5) It would be nice if you could visualize the affordances learned by the model, by showing visual attention maps. As you always have a single object on the scene, would your approach also work if you have two or three tools on the scene and gave a language instruction to manipulate one of them?
5) How crucial is the meta learning part for adaptation? It's hard to tell what part needs adaptation, the linguistic part, the visual part or the control policy. Using  also a pretrained visual encoder might help with the generalization.

[0] What Matters in Language Conditioned Imitation Learning over Unstructured Data
[1] CLIPort: What and Where Pathways for Robotic Manipulation
[2] Grounding Language with Visual Affordances over Unstructured Data

---

### Official Review · Reviewer_q9mS · 2022-11-13
**Interesting study. A greater diversity of objects, and low-level policies/motion sequences along with a more rigorous analysis showing where the benefits of additional linguistic information add on would be nice.**

**Rating:** 6
**Confidence:** 4

**Review:**

The paper studies how linguistic information about the affordances of objects can aid the adaption of policies to new tools. This is done by meta-learning policies that can efficiently adapt to new tools given their corresponding text descriptions.

Strengths:
1. The idea is interesting and novel.
2. The paper is well written, and the video demonstration puts across the point of the paper.

Weaknesses
1. Unclear what portion of the policies are _really_ getting influenced by additional linguistic information. Is it where the objects are being grasped from? Is it the policy executed after grasping?
2. The sequences of actions that follow grasping across the tasks look almost identical. The success on several of these tasks might even be indifferent to where the correct grasp is performed. More diverse motions are required to understand if the benefits of any additional linguistic information can help low-level decisions and control.
3. A rigorous set of baselines are required. The lack of significant variance in the executed motion across different objects results in questioning whether the experiments even set out to solve the question initially proposed.

---

### Decision · Program_Chairs · 2022-11-15

Accept (Poster)